# Developing a Systematic Diagnostic Model for Integrated Agricultural Supply and Processing Systems

**Mduduzi Innocent Shongwe [1,*][ID], Carel Nicolaas Bezuidenhout [1], Milindi Sylver Sibomana [1], Tilahun Seyoum Workneh [1], Shamim Bodhanya [2] and Vukile Vinah Dlamini [3]**

[1] Department of Bioresources Engineering, University of KwaZulu-Natal, Private Bag X01 Scottsville, Pietermaritzburg 3209, South Africa; carel.bezuidenhout@yahoo.com (C.N.B.); sylvermilindi@gmail.com (M.S.S.); seyoum@ukzn.ac.za (T.S.W.)

[2] Graduate School of Business and Leadership, University of KwaZulu-Natal, Private Bag X54001, Durban 3600, South Africa; bodhanyas1@ukzn.ac.za

[3] Department of Consumer and Food Science, University of Pretoria, Private Bag X20, Hatfield 0028, South Africa; vukkyle@hotmail.com

\* Correspondence: mduduzi_shongwe@yahoo.co.uk

**Abstract:** Despite all the innovative research in agriculture, technology adoption in integrated agricultural supply and processing systems (IASPS) remains a challenge. This is attributed to the complex nature of IASPS and the continued lack of a holistic view towards most of the interventions into the systems. To make sense of issues that affect IASPS, it is important to recognise that most issues within these systems do not exist in isolation, but are imbedded within complex interrelationships. This research developed and demonstrated a systematic diagnostic model that could be used to locate high leverage intervention points within IASPS and to make predictions about the systems behaviour. A meta-analysis was conducted to test the evidence of the interlinkages between IASPS domains and to compare the strength of these relationships. The model revealed that the collaboration, structure and information sharing domains had a higher direct leverage over the other IASPS domains as these were associated with a larger number of interlinkages. Additionally, collaboration and structure provided dynamic leverage as these domains were part of feedback loops. In terms of the potency, collaboration was highly correlated to culture compared to the other domains, viz., information sharing, coercive power and transaction costs.

**Keywords:** adoption; interlinkages; meta-analysis; supply systems; systems thinking

## 1. Introduction

Innovation is a valuable asset that gives supply chains the competitive edge. It increases productivity and the quality of service [1]. Ham and Johnson are of the view that interorganisational innovation increases the level of cross organisational interoperability and integration [2]. Regardless of the overwhelming benefits of innovation, the adoption of technology in supply chains remains a major concern [3]. The adoption of technologies in integrated agricultural supply and processing systems (IASPS) is no exception as nonadoption is widely reported in research [4–6]. According to McCown, the adoption of technologies in IASPS and, in particular, integrated sugarcane supply and processing systems (ISSPS), has been relatively slow especially when compared to other industries such as electronics and automotive [7]. Higgins et al. attribute the slow adoption to system complexity [6]. Agricultural systems are characterised by interactions between, amongst others, multiple stakeholders,

the environment, land use planning and finances. As such, agricultural systems are viewed as complex socioecological [8] or sociotechnical [9] systems.

Multiple causes underlie most of the dynamics of agriculture related problems. As such, interventions into IASPS require diverse and content-specific solutions. However, most research in agricultural systems still focus on short-term rather than long-term solutions. These short-term interventions however, consider only some parts of the system and in most cases view IASPS as technical systems [6,8]. Integrated agricultural supply and processing systems exhibit several complex system's characteristics such as nonlinear dynamics, feedback, emergence, delay, and counterintuitive behaviour [10]. Solutions to such contexts as a consequence, rely on the interactions between the various system components rather than on isolated components [9]. A systems thinking approach as such, is required to unlock and to understand the adoption of technologies in IASPS. Technology adoption in IASPS therefore, is more possible only when all domains are considered simultaneously [11].

A range of models are used to study technology adoption and amongst the mostly used is the diffusion of innovation framework [12] and the technology acceptance model [13]. The diffusion of innovation framework (DOI) covers five characteristics of technological innovations, viz., relative advantage, compatibility, complexity, triability and observability. According to Hsu et al., the DOI can explain 49–87% of the variance in adoption rates [14]. The technology acceptance model (TAM), on the other hand, comprises two critical factors: perceived usefulness and ease of use. The DOI and TAM are, however, primarily based on an individual's acceptance behaviour. The adoption of technologies by organisations is nonetheless different from that of individuals as it involves multiple decision-makers. The application of DOI and TAM in such contexts therefore, is not sufficient. The technology, organisation and environment context model (TOE) by Tornatzky and Fleischer is a widely accepted approach at organisational level [15]. According to Tornatzky and Fleischer, an organisation's decision to adopt does not only lie with the characteristics of the technology itself but also with the organisational capability and the environmental context [15]. Organisational context as used in the model refers to the characteristics and resources of the entity. More broadly, organisational factors, such as structure [16], strategy [17], culture [18] and economics [19], have all been studied to establish their role in the adoption of technologies. The environmental context on the other hand, represents the setting within which an organisation operates.

Technology adoption in supply chains introduces other dimensions to those of organisations. Since supply chain initiatives impact on operational routines and relational structures there is an obvious need to consider interactions between interorganisational factors. Interorganisational interactions induce and/or increase uncertainty. These interorganisational factors are widely explained through institutional theory [20]. Institutional theory is based on the premise that organisational changes are not driven by intraorganisational and technological criteria only but also the pressure to conform. Therefore, organisations sharing the same environment become isomorphic with each other [21]. Institutional theory as such, is widely applied in interorganisational adoption research, whether in isolation or in combination with other factors and/or models. Table 1 shows some adoption factors that have been studied across supply chains.

**Table 1.** Adoption factors associated with supply chains.

| Researchers | Adoption Factors | | | | | | | | | |
|---|---|---|---|---|---|---|---|---|---|---|
| | **Physical** | **Collaboration** | **Culture** | **Economics** | **Environment** | **Strategy** | **Information Sharing** | **Power** | **Structures** | **History** |
| Chatterjee et al. [22] | | | √ | | | √ | | | √ | |
| Hsu et al. [14] | | | | √ | √ | | | √ | | |
| Seymour et al. [23] | √ | | √ | √ | √ | √ | √ | | √ | |
| Chong et al. [24] | | √ | | √ | | | √ | | | |
| Ranganathan and Jha [25] | | | | | √ | | √ | | √ | |
| Pang and Bunker [26] | | √ | | | √ | | | √ | | |
| Johnston and Gregor [27] | √ | √ | √ | √ | √ | | | √ | | |
| Patterson et al. [28] | | | | √ | √ | | | √ | √ | √ |
| Matopoulos et al. [29] | | √ | | | | √ | | √ | √ | |
| Bezuidenhout et al. [5] | √ | √ | √ | √ | √ | √ | √ | √ | √ | √ |
| Schut et al. [30] | √ | | √ | √ | | | | √ | √ | |

As indicated in Table 1, different supply chain adoption issues often align themselves with different and/or a combinations of factors or domains. Johnston and Gregor [22] and Patterson et al. [23] provide a more general list of factors that affect the adoption of technologies in supply chains. Seymour et al. [24] considered factors that affected adoption in a container supply chain. Schut et al. [25] and Bezuidenhout et al. [5] conceptualised factors from an integrated agricultural systems' view. Also, as is clear from Table 1, numerous factors have been studied more often than others and amongst the most researched are the physical factors, collaboration, culture, economics, strategy, information sharing, power, structure and the environment. The history domain is not common among researchers. According to Bezuidenhout and Bodhanya, the description of history as an adoption factor is not clear and often lacks consistency [26].

The "factor approach" (TOE model and institutional theory), however, is static and tends to view adopters as being passive [27]. These approaches do not capture the complex and dynamic nature of interorganisational linkages introduced at adoption [28]. Hence, a processual approach is widely advocated for [29]. A processual approach views system's behaviour as being emergent and accordingly captures the interplay of interactions between an individual firm, the industry and the environment it operates on. Researchers have studied numerous interlinkages between some of the many supply chain adoption factors. For example, Defee and Stank studied the interlinkages between strategy, environmental uncertainty and supply chain structure [30]. A study by Abosag explored interlinkages between economics, culture, information sharing and collaboration [31]. Kang et al. studied the relationship between power, collaboration, culture and communication [32].

This research developed and demonstrated a systematic diagnostic model that explores and compares the interlinkages between the many IASPS adoption domains, viz., biophysical, collaboration, culture, economics, environment, future strategy, information sharing, political forces and supply chain structure. Knowledge of these interlinkages is important given that technological adoption in supply chains requires a comprehensive approach As such, it is vital to a have a model within which to work and from which testable hypotheses could be drawn. The model will be used to locate high leverage intervention points within IASPS and to make predictions about the systems' behaviour. To the researchers' knowledge, there are currently no studies that have comprehensively considered the interlinkages between all of these IASPS within a single intervention. The use of the model is therefore expected to improve the efficiency of systemic diagnosis of issues within IASPS. The remainder of this section describes the IASPS adoption domains in brief and highlights their key antecedents, consequences and barriers. The interlinkages between the domains are further identified and these informed the formulation of several hypotheses.

*Adoption Domains and Hypotheses*

The study conceptualised IASPS as a sum of the biophysical domain, collaboration, culture, economics, environment, future strategy, information sharing, political forces and supply chain structure [5]. Structure describes the tasks, authority and coordination mechanisms across the distinct parts that form a supply system. There are many structural dimensions proposed in literature but integration and communication are widely used in supply chains [33]. Integration refers to the alignment and coordination of processes and functions across the supply system. Steven and Johnson recognise three forms of supply chain integration, viz., information integration, coordination and organisational linkages [34].

The environment domain is the context within which the supply chain exists. It is multidimensional and comprises both macro and micro factors. The multidimensionality of the environment brings about uncertainty into the supply chain. This uncertainty unfortunately gives rise to adaptation and evaluation challenges. As such environmental uncertainty is the main driver for seeking flexibility [35]. Scott and Davis classify environmental uncertainty into complexity and dynamism [36]. Environmental dynamism is the rate of change and turnover in the environment

whilst complexity represents the diversity of and/or interdependence between numerous issues that the system has to cope with.

Future strategy monitors the environment for threats and opportunities. As such, strategy determines supply chain goals and configurations. There are two generic supply chain strategies, viz., lean and agile. A lean strategy focuses on increasing efficiency through the elimination of waste. Agile strategies in contrast, are founded on structures that are capable of competing in highly dynamic and unpredictable environments. There are four concepts inherent to agility, viz., flexibility, responsiveness, competency and speed. Most agricultural systems are characterised by both lean and agile principles [37]. The ISSPS for example, require lean principles to adapt to a commodity-type market downstream whilst upstream they require agile strategies to deal with multiple stakeholders and high production risks [38].

Several researchers advocate for the adoption of the structure–strategystructure–strategy–performance paradigm within the supply chain context [30,39,40]. The structure–strategystructure–strategy–performance paradigm (SSP) posits that a firm's strategy drives its structure and performance. Furthermore, the SSP put forward that the structure–strategy relationship is contingent to the external environmental. According to Effendi and Arafin, the relationship between structure and strategy is inextricably reciprocal [41]. Structure should be compatible with strategy otherwise strategy formulation and implementation will be constrained. Agile supply chains require smooth coordination and integration of functions across supply chain members. According to Tse et al., agility moderates the effect of integration on performance [42]. Integration of activities across the supply chain improves the ability to respond to volatile markets. According to Far et al., the integration of IT systems between partners can increase the rate of agility which could allow partners to exploit interorganisational collaborations [43]. Empirical findings from a multicase study by Ngai et al. found that integration has a positive and significant effect on agility [44]. Similarly, Cagliano et al. found an association between integration and the lean supply chain strategy [45]. Therefore, the following hypothesis was made.

**Hypothesis 1 (H1).** *Supply chain structure is correlated to supply chain strategy.*

The environment is either exogenous to the SSP or have a direct relationship [46]. According to Decheng and Yu, environmental uncertainty is positively correlated to supply chain integration [47]. Integration, through improved responsiveness, mitigates the impact of environmental uncertainty on performance [48]. As such, environmental uncertainty would lead a firm to reshape its business model to pursue the best operational practice. According to Salvato and Vassolo, dynamic environments are mostly associated with higher levels of integration [49]. In a volatile demand market, firms should enhance supplier, customer and internal integration to increase their agility [50]. A study by Chi et al. [51] found a positive and statistically significant path between environmental dynamism and supply chain structures. Similarly, Wu et al. found demand uncertainty to positively moderate the relationship between customer integration and green product innovation [50]. Therefore, the following hypothesis was made.

**Hypothesis 2 (H2).** *Environment is associated with supply chain structure.*

As partly proposed by the SSP debate, sustainable competitive advantage is achieved through a fit between the environment and both the structure and strategy [46,48]. More so, highly uncertain environments are characterised by agile strategies whilst lean strategies are common among low uncertainty contexts [52]. Ambe points out that agile strategies are more appropriate in turbulent environment as they responds quickest to dynamic conditions [53]. In contrast, lean strategies perform better in a stable, predictable context. Empirical evidence by Gligor et al. shows a positive and significant association between agility and customer uncertainty [54]. A study by Um found supply chain agility to be positively related to a differentiation strategy in a high-level customisation environment [55]. Therefore, the following hypothesis was made.

**Hypothesis 3 (H3).** *The environmental domain is correlated to strategy.*

The information sharing domain describes the extent to which critical and proprietary information is communicated between supply chain partners. It refers to the act of capturing and dissemination. Restricted information flow not only obstructs the ability to prepare for sudden changes but impedes adaptation to environmental changes. Information sharing as such, is the heart [56] and nerve centre [57] of supply chain collaboration. Khurana et al. recognise four broad barriers to information sharing in supply chains, viz., managerial, technological, individual characteristics and sociocultural factors [58].

Supply chain collaboration is when two or more independent supply chain members work mutually together to arrange and to execute operations with more prominent accomplishment than when acting in isolation. Kumar et al. describe collaboration as interfirm linkages where members share information, resources and risk to accomplish mutual objectives [59]. The key antecedents to supply chain collaboration are information sharing, decision synchronisation and incentive alignment [60]. Collaboration is often defined by trust, commitment, cooperation and coordination. According to Weaver, collaboration marks the final phase of the C3 model (coordination, cooperation and collaboration), meaning that collaboration requires coordination and cooperation as prerequisites [61].

Supply chain integration leads to timely and accurate information sharing. Consequently, integration improves communication channels between supply chain members. Research findings by Sahin and Robinson show a positive correlation between logistics integration and information sharing [62]. A study by Mansoori et al. also found that supply chain integration has a positive and significant impact on information sharing [63]. Therefore, the following hypothesis was made.

**Hypothesis 4 (H4).** *Supply chain structure is correlated to information sharing.*

Cooperation and trust are reciprocal processes depending on and fostering each other [64]. Soosay and Hyland state that trust, cooperation and commitment are a dynamic process where partners constantly evaluate their decisions on whether to continue with a particular relationship or not [65]. A study by Hardman et al. on the South African apple value chain found that trust leads to cooperation and in turn, commitment [66]. Masuku and Kirsten came to the same conclusion in a study on ISSPS [67]. A lack of trust is an obstacle to information sharing. Conversely, high levels of trust reduce the fear of information disclosure.

Zand's dynamic trust model portrays the trust–information sharing relationship as a reinforcing spiral [68]. When a relationship is based on mistrust, this spiral deteriorates into decreased information sharing and subsequently, reduced trust. A study by Nyaga et al. found trust to be positively correlated to information sharing [69]. Kim and Lee discovered an increase in information sharing capabilities as a consequence of increased trust levels [70]. Therefore, the following hypothesis was made.

**Hypothesis 5 (H5).** *Collaboration is correlated to information sharing.*

The biophysical or material domain refers to the network structure of physical equipment and processes used to enable value addition. It includes raw materials, work-in-process inventory and finished products. Large parts of agricultural supply chains deal with biologically-active produce/products, hence the use of biophysical over material flow domain throughout this article. An efficient physical flow system guarantees on-time delivery which in turn, ensures that inventory levels (and costs) are kept minimal. Sugarcane supply synchronisation is considered critical within ISSPS as it promotes capacity utilisation, mitigates material handling risks, minimises stockpiling and reduces sugarcane deterioration [26].

According to Kaipia et al., information sharing leads to improved inventory management, higher sales and to a better understanding of demand [71]. It enables supply chain members to plan properly and to avoid inventory bottlenecks. The sharing of inventory information improves order

replenishment, safety stock placement and trans-shipment. In vendor-managed systems, suppliers are continuously updated on inventory levels and sales data via electronic data interchange systems and replenishments are often automatically generated once the inventory drops below certain levels. Therefore, the following hypothesis was made.

**Hypothesis 6 (H6).** *Information sharing is correlated to the biophysical domain.*

The value of stock in ISSPS is often outweighed by rapid sugarcane deterioration and, hence, stockpiling only occurs on the basis of inconsistent supply and demand. Moreover, downstream inventory performance is often positively related to an increase in market share, sales and profit [72]. Empirical findings by Shah and Shin show inventory levels to have a direct link to financial performance [73]. A study by Agus and Shukri Hajinoor found a positive correlation between inventory control and both return on sales and profitability [74]. Research conducted in sugar manufacturing firms by Lwiki et al. concluded that there is a positive correlation between inventory control and return on equity [75]. Therefore, the following hypothesis was made.

**Hypothesis 7 (H7).** *The biophysical domain is correlated to the economic domain.*

Various researchers have studied the effect of supply chain collaboration on firm performance and mostly concluded that higher levels of collaboration leads to better performance [76–78]. Improved cooperation and coordination improves on-time delivery and greater responsiveness. Collaboration helps partners to share risks and to reduce transaction costs. According to Jiang et al., trust is a substitute for formal contracts as it reduces relational risk [79]. In areas of high trust, transacting partners spend less time on ex ante contracts because they are confident that partners will not be opportunistic [80]. In their study, Zaheer et al. found a negative relationship between interorganisational trust and negotiation costs [81]. Research findings by Um and Kim indicated that the collaboration–transaction cost relationship is moderated by contractual and relational governance mechanisms [82]. A shared sense of identity motivates partners to seek the best interests of a transaction. Similarly, contracts allow parties to work as promised.

The need for decision synchronisation in supply chains lies with the potential increase in "collective pay-off" in terms of overall profits and costs [83]. Accordingly, Pol and Inamdar state that vendor managed inventory (VMI) reduces inventory buffers and the need for extra capacity [84]. Through VMI suppliers are able to coordinate transport and to make more efficient route planning [85]. Findings by Irungu and Wanjau indicated that VMI promotes faster inventory turns by reducing carrying costs, inventory holding and product spoilage [86]. In ISSPS, grower consortiums are often touted as a transaction cost reduction strategy [87]. Therefore, the following hypothesis was made.

**Hypothesis 8 (H8).** *Collaboration is correlated to the economics domain.*

Supply chain culture describes the patterns of shared values, beliefs, assumptions and behaviour among supply chain members. Culture facilitates interorganisational learning and is often viewed as a direct precursor to trust and commitment [88]. According to Schein, culture manifests itself at three levels, viz., artefacts, espoused values and underlying assumptions [89]. Artefacts are the visible aspects and consist of the physical and social settings. Values represent conscious, affective desires whilst assumptions embody an unconscious aspect.

Culture is a direct precursor to trust and commitment [88]. According to Zhang et al., the relationship between trust and shared values is reciprocal with shared values helping to create a relationship built on trust, and trust serving to maintain and to express those values [90]. Accordingly, Morgan and Hunt noted that when exchange partners share values they become more committed to a relationship [91]. Based on the notion that culture promotes behavioural consistency, Bouachouch and

Mamad argue that culture facilitates coordination [92]. Empirical findings from Urbancova indicate that culture affects both cooperation and trust [93]. Therefore, the following hypothesis was made.

**Hypothesis 9 (H9).** *Culture is correlated to collaboration.*

Political forces describe those actions that influence resources but are not part of one's formal role [94]. These are an important aspect of deciding "what does or does not get done" [95]. Accordingly, politically-oriented behaviour manifests itself through the exercise of power as power serves as a mechanism for achieving compliance. According to Maloni and Benton, power is either mediated or nonmediated [96]. Mediated power describes those bases that are deliberately engaged to guide response, e.g., reward power, coercive and legitimate power. In contrast, nonmediated power describes those forms that are more relational and positive in orientation, e.g., expert and referent power. Power is also conceptualised from a resource-dependency perspective where supply chain partners are viewed as interdependent entities seeking to manage uncertainty [97]. Another approach to power is derived from transaction cost economics where partnerships are motivated by self-interests [98].

According to Belaya and Hanf, coercive power is negatively correlated to collaboration [99]. Leonidou et al. are of the view that continuous use of coercive power between partners degrades trust [100]. Empirical research by Maloni and Benton show a negative relationship between coercive power and cooperation [96]. Another study by Cheng et al. found that the coercive bases of power increase conflict [101]. Therefore, the following hypothesis was made.

**Hypothesis 10 (H10).** *Political forces are related to collaboration.*

The article is organised into four sections. The next section presents the methods undertaken to conduct the research. This is followed by a section on results and discussion. The conclusion section follows thereafter.

## 2. Methods

A meta-analysis was conducted to test the evidence of the hypotheses developed in Section 1 (Introduction) and to compare the strength of the various interlinkages. This was essential since most of the research on the interlinkages is drawn from multiple disciplines where the domains are often operationalised differently. Meta-analysis provides a systematic statistical analysis of independent studies. Results from meta-analysis are as such, better than those from single studies considering that meta-analysis integrates diverse sets of population. This increases precision around the overall mean effect and reduces the sampling error [102]. Meta-analysis as used in this study allowed the researchers to build a comprehensive model that have not been examined in any individual primary study.

Peer-reviewed articles published from years 2000 to 2017 were consulted for the meta-analysis. Although the emphasis was on agricultural supply chains, articles from other supply systems were considered for Hypothesis 10. This was due to a shortage of empirical research specific to IASPS (not enough to warrant meta-analysis) for this particular hypothesis. The collaboration–political force relationship in this research therefore, is viewed from a domain perspective rather than a meta-analysis specific to IASPS. The search for relevant articles began with a keyword search using the domains and/or domain dimensions. Academic search engines including Web of Science, EBSCO and ProQuest were used to identify relevant studies. A manual search of journals was also conducted. To be considered in the meta-analysis, articles had to report on an effective size statistic on a relationship between any of the domains. The Pearson correlation coefficient ($r$) served as the effect size metric and studies that reported other metrics (e.g., F-test and T-test) were converted to $r$ using appropriate formulas (refer to [103]). After a thorough "sifting" exercise, one-hundred-and-thirty-five studies were included in the meta-analysis.

Each effect size was corrected for sources of error (sampling error, attenuation and reliability) using a weighted average reliability value of the sampled studies [104]. Corrected correlation coefficients ($r_c$)

for each hypothesis were subsequently computed. Lastly, Cochran's Q-tests and credibility intervals (CV) were computed for each hypothesis to quantify heterogeneity. A null hypothesis in a Q-test assumes that all studies come from the same population. A significant Q-test as such, indicates that effect sizes are heterogeneous.

Schmidt and Hunter discourage the use of the Q-test in isolation especially when the number of studies considered is less than six and/or when the average sample size is less than thirty [102]. It is argued that in such cases, the Q-test tends to accept the null hypothesis even though with an unknown type II error rate [105]. The Q-test in addition, tends to also reject the null hypothesis when the number of studies is large. The sample sizes of the studies considered in this research ranged from 4 to 1174. Hence, a Q-test may not have been sufficiently able to accurately reflect heterogeneity. Credibility intervals were therefore, computed alongside the Q-test. Credibility intervals provide an estimate of variability in the distribution of correlation values. A large credibility interval or that which includes zero assumes heterogeneity and indicates the presence of moderators [106].

A case study was thereafter conducted at a sugarcane milling area in Swaziland to demonstrate the diagnostic model. Integrated sugarcane supply and processing systems are complex systems characterised by messy and/or tame problems. Messy problems are a class of social problems where there are differences in opinions about the problem and/or the question of whether there a problem exists or not. These types of problems are continually evolving, have many causal levels and have no single solution. On the contrary, tame contexts are well-defined and can be solved linearly using reductionist and/or sequential techniques. To make sense of the issues that affect ISSPS, it is important to recognise that such contexts should be approached using systematic social processes. These interpretive approaches understand reality as expressed by subjective experiences of individuals.

The study was an exploratory survey where the objectives were to (1) identify issues that constrained productivity in the area and to (2) prioritise an area of further research focus from the identified issues. A purposive sample of seven stakeholders was selected for telephone interviews. These respondents were the sugarcane laboratory manager, sugarcane supply manager, extension services manager, factory manager, harvesting haulers (2) and a grower. The respondents represented stakeholders that had been actively involved within the milling area for at least two consecutive years. To the researchers' knowledge, these respondents were highly involved with systemic issues within the area and as such, provided representative viewpoints from their specific profiles. Rather than size, the sample was guided by adequacy, accessibility and availability of the stakeholders.

The individual interviews were recorded in an audio tape. The audio tape recordings were transcribed and modelled into a rich picture diagram. Furthermore, constant comparative analysis was used to analyse the interviews. Rich pictures are graphical modelling tools from soft systems methodology (SSM). Soft systems methodology is widely used to unlock social complexity through structuring and interpretation of messy problems. The SSM considers problems as constructs of the mind, defined by stakeholder worldviews. Its interpretive nature means that there is no definite problem definition but perspectives. Therefore, soft systems methodology models do not describe the real world but generate "holonic ideal type" of human activity system's behaviour under a certain perspective. Checkland and Poulter describe SSM as a four-stage learning cycle [95]. These stages are, namely, perceived real-world problematic situation, purposeful activity models and structured discussions about change and action to improve (refer to Checkland and Poulter [95] for more in-depth discussion). The SSM have a set of tools that help researchers carry out these stages, viz., rich pictures, conceptual models, CATWOE and formal system models. Since the survey was diagnostic in nature, only the perceived real-world problematic situation stage was considered, hence, the use of rich pictures. The use of tools outside a methodology is premised on multimethodology which is the creative combination of methodologies or parts thereof within a single intervention [4]. Rich pictures provide a detailed, high-grained view of a problem context. The drawing of rich pictures however, does not have a specific format or language. This makes interpretation third party interpretation difficult.

The rich picture was thereafter presented to stakeholders at a report back meeting that was held at the milling area offices. The meeting was attended by four of the earlier interviewees', viz., sugarcane laboratory manager, sugarcane supply manager, extension services manager and the factory manager. The objectives of the meeting were to present the findings to the stakeholders and to collect data on issues that may have been missed by the interviews. More importantly, the meeting served as a platform to facilitate a shared problem definition and commitment for further action from as this could not be attained through the telephone interviews. The use of rich pictures in open discussions ensured findings credibility, dependability and conformability. The discussions enabled member checks and further allowed data collection. According to Anney, member checks improve credibility and transferability of findings [107]. The use of interviews and open discussions in the research further facilitated methodological triangulation. Triangulation as stated by Treharne and Riggs, increases credibility and conformability [108].

Constant comparative analysis (CCA) is a grounded theory technique that uncovers patterns, themes and categories important to a social reality [109]. According to Charmaz, the ultimate goal of CCA is to link and integrate categories in such a way that variation is captured by an emerging theory [110]. As such, the researchers used the interview transcripts to categorise the issues that constrained productivity in the area along the IASPS domains. Causal relations between the issues were further established through a review of previous studies.

There were however, possible limitations to the case study. Despite the fact that the researchers attempted to obtain a representative sample, there could be bias because only seven interviews out of potentially hundreds. Also, stakeholder representation at the meeting was skewed towards the factory (laboratory manager, sugarcane supply manager and factory manager). Furthermore, the use of open discussions could be influenced by dominant individuals and that may have introduced some bias.

## 3. Results and Discussion

The results and discussion section is divided into two sections. The systematic diagnostic model is presented in the Section 3.1 whilst the case study is presented in Section 3.2.

### 3.1. Systematic Diagnostic Model for IASPS

The open nature of the IASPS domains meant that it was difficult to cover all domain dimensions within the meta-analysis. It is for this reason that only a few dimensions from each domain were selected for the research. Dimensions, as used in this article, refer to the various constructs or forms that make up a domain, for example, structure can be formalisation, centralisation, complexity and/or integration. Table 2 shows results from the meta-analysis and as indicated the $r_c$ were ranked according to the correlation threshold scale (SE) by [111]. According to Cohen, correlations between 0.10 and 0.3 are regarded as small (S) [111]. Accordingly, correlations between 0.30 and 0.50 are categorised as medium (M) whilst those above 0.50 are considered large (L). Cohen further suggested that any correlation smaller than 0.10 is trivial [111]. Most researchers, however, are critical of the SE and many argue that the effectiveness of any intervention can only be interpreted within the context of the research domain that is being evaluated [112]. Also indicated in Table 2 is the number of independent samples consulted (k) and the overall sample size (N).

**Table 2.** A meta-analysis of agricultural supply and processing domains.

| Hypothesis | k | N | $r_o$ | $r_c$ | SE | 95% CV | | Q |
| --- | --- | --- | --- | --- | --- | --- | --- | --- |
| | | | | | | Upper | Lower | |
| H1 (Structure–strategy) [1] | 10 | 1914 | 0.321 | 0.322 | M | 0.374 | 0.272 | 8.84 |
| H2 (Structure–environment) [2] | 11 | 1894 | 0.062 | 0.054 | Trivial | 0.069 | 0.038 | 14.51 |
| H3 (Strategy–environment) [3] | 15 | 2514 | 0.310 | 0.295 | S | 0.350 | 0.239 | 11.81 |
| H4 (Structure–information sharing) [4] | 10 | 2298 | 0.643 | 0.594 | L | 0.669 | 0.519 | 9.85 |
| H5 (Collaboration–information sharing) | 21 | 6810 | 0.530 | 0.468 | M | 0.540 | 0.396 | 15.00 |
| H6 (Information sharing–biophysical) [5] | 11 | 2029 | 0.336 | 0.372 | M | 0.434 | 0.309 | 8.68 |
| H7 (Biophysical–economic) [6] | 10 | 382 | 0.822 | 0.728 | L | 0.837 | 0.618 | 10.78 |
| H8 (Collaboration–economics) [7] | 11 | 2935 | −0.103 | −0.145 | S | −0.207 | −0.019 | 9.27 |
| H9 (Culture–collaboration) | 17 | 4776 | 0.595 | 0.545 | L | 0.619 | 0.469 | 20.07 |
| H10 (Political forces–collaboration) [8] | 19 | 4283 | −0.671 | −0.313 | M | −0.133 | −0.494 | 4.41 * |

[1] Integration–agile strategy; [2] Integration–environmental uncertainty; [3] Flexibility–environmental uncertainty; [4] Integration–information sharing; [5] Inventory levels–information sharing; [6] Inventory–return on sales; [7] Trust–transaction costs; [8] Trust-mediated power. * $p < 0.05$.

The Q-test was statistically insignificant ($p < 0.05$) for all hypotheses except for Hypothesis 10 (political forces–collaboration), indicating that the effect sizes were homogeneous. This was further supported by the credibility intervals as all the values (Hypothesis 1 to Hypothesis 9) excluded zero. The statistically significant Q-test for Hypothesis 10 indicates heterogeneity. The CV for Hypothesis 10 was fairly wide ($-0.133$ to $-0.494$), suggesting that moderators may have existed. Various researchers argue that the relationship between trust and coercive power is moderated by commitment [113,114]. Jain et al. are of the view that the effect of coercive power on trust decreases with an increase in affective commitment [114].

According to the SE (Table 2), Hypothesis 4, 7 and 9 were large, while Hypothesis 1, 5, 6 and 10 were categorised as medium. Hypothesis 2 was classified as trivial. This research, however, uses the SE only as a guide. Hence, it does not view Hypothesis 2 as insubstantial. According to the SE, the strategy-environment and the collaboration–economics average effect sizes were classified as small. A stronger correlation increases the predictive worth of an interaction hence, is the case for Hypothesis 4, 7 and 9. This implies that the knowledge of either factor can be used to predict the behaviour of the other. Referring to Figure 1, information sharing is seen to be more predictive of structure ($r_c = 0.594$) compared to collaboration ($r_c = 0.468$) and the biophysical domain ($r_c = 0.373$). Based on the potency of the interlinkages it is recommended that in cases where information sharing is viewed as a constraint, decision-makers should comprehensively consider the role of structure, collaboration and then the biophysical domain, respectively.

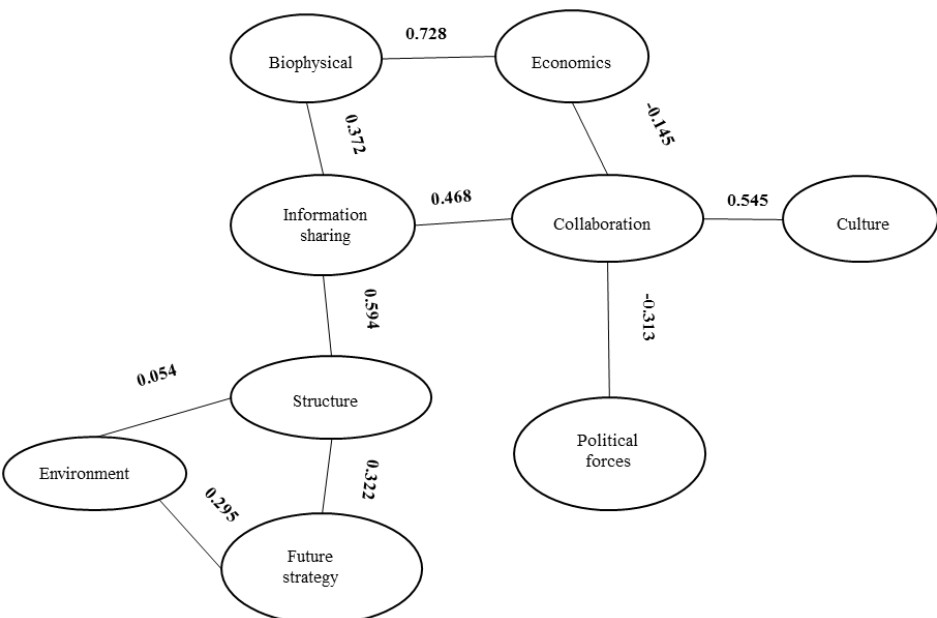

**Figure 1.** Systematic diagnostic model for integrated agricultural supply and processing systems (IASPS).

The correlation value between information sharing and collaboration ($r_c = 0.468$) and that between information sharing and structure ($r_c = 0.594$) were compatible with the research findings of Kalyar et al. [115]. The study by Kalyar et al. concluded that legal protection ($r = 0.463$) and trust ($r = 0.444$) were mostly predictive of information sharing [115]. Based on the potency of interlinkages, collaboration was found to be strongly correlated to culture ($r_c = 0.545$) compared to its relationship with information sharing ($r_c = 0.468$), coercive power ($r_c = -0.313$) and transaction costs ($r_c = -0.145$). These findings are consistent with those of Fawcett et al. that revealed culture and information sharing to be the most potent barriers to supply chain collaboration [116]. Diagnostically, a low collaboration index may imply the overuse of coercive power, mismatched values, information sharing issues and/or economic issues.

As can be seen in Figure 1, information sharing and structure are the most central domains in IASPS directly influencing four (culture, economics, information sharing and politics), three (biophysical, collaboration and structure) and three (environment, information sharing and strategy) domains, respectively. This means that these domains hold a relatively higher direct leverage in IASPS. Furthermore, the relationship between structure, environment and strategy forms a feedback loop (positive loop considering the domain constructs considered in the meta-analysis). Positive loops are self-reinforcing whilst negative loops exhibit a goal-seeking behaviour. According to Nguyen and Bosch, feedback loops are an important source of dynamic leverage [117]. Dynamic leverage focuses on cause-and-effect relationships that feedback over time. As much, dynamic leverage minimises the amount of initial effort required to set a system moving and the amount of maintenance forces required to keep feedback structures in place. For example, external integration (structure) could be used to leverage environmental uncertainty. With higher levels of integration, supply chain partners have access to more current and accurate information. This allows tight coordination and ensures that the supply chain is more flexible (strategy) to environmental changes. Information sharing, collaboration, economics and the biophysical domain also form a feedback loop. Collaboration could, for example, be used to leverage issues in the biophysical domain. An increase in trust within the supply chain could increase the level of information sharing and by so doing, inventory data could become more accessible which may improve coordination and consequently reduce transactional costs. The loops in Figure 1 imply that any intervention into IASPS should strive to simultaneously consider collaboration, information sharing and structure as these have a higher leverage (direct and dynamic).

The high leverage position of information sharing within IASPS is visible from Figure 1. Through information sharing, structure links strategic factors (environment and strategy) to operational domains (collaboration and biophysical). Information sharing further affects both feedback loops on Figure 1. The relationship between information sharing and the collaboration–economics–biophysical loop provides higher leverage compared to that with the structure–strategy–environment loop. As indicated, information sharing forms part of the collaboration–economics–biophysical loop whilst it acts as an exogenous factor towards the structure–strategy–environment loop, only directly affecting structure. These findings support Lofti et al.'s argument that information sharing is at the heart of supply chain management [56].

Although important, the influence of culture and political forces on the overall domains provide low leverage. As indicated in Figure 1, culture and political forces are only linked to the collaboration domain. Moreover, the correlation between collaboration and culture was large especially when compared to that between collaboration and economics (small), political forces (medium) and information sharing (medium).

After obtaining the meta-analysis results as indicated in Table 2, funnel plots (not shown here) were created to investigate the presence of publication bias. Publication bias leads to a nonrepresentative database that overestimates true effect sizes. According to Ahmed et al., research findings that are statistically significant have a higher chance of being published compared to nonsignificant research [118]. After visual inspection of each funnel plot, it was concluded that no meaningful publication bias existed in the meta-analysis.

### 3.2. Case Study

Results from the interviews with stakeholders are as shown on the rich picture (Figure 2). The rich picture was presented at the report back meeting with the stakeholders. The issues that constrained productivity in the area were environmental (rainfall), biophysical (farm roads, factory stops and sugarcane quality), structural (harvesting contracts and irrigation water), political (unfulfilled harvesting and haulage schedules) and cultural (labour unrest, vehicle labelling, consignment numbers and field numbers). Rainfall was widely viewed as the main constraint in the area as reported by the growers, the extension services manager and the factory manager. According to the growers and the extension services officer, excessive rainfall was a serious problem especially at harvesting. Under

extreme rainfall conditions, it became difficult for haulers to manoeuvre farm roads. Continuous wet weather further reduced the ability to preharvest burn sugarcane and increased the chances of soil contamination in the sugarcane delivered to the factory. The environmental domain, strategy and the structural domain form a causal loop (Figure 1), hence, interventions on the issues of rainfall (environment) in the area could be considered along the length of the milling season (structure) and the flexibility of factory operations (strategic). This arrangement is common in Swaziland, as reported by Mhlanga-Ndlovu and Nhamo, that, in the past, extreme rainfall had led to an adjustment to the length of the milling season by 4 to 6 weeks of the normal period [119]. Accordingly, Bezuidenhout advocates for a "controlled system variability" principle towards mill capacity utilisation in order to accommodate unexpected events such as extreme rainfall [38].

Also indicated in Figure 2 is the issue of unreliable sugarcane supply. There are, however, other issues in the rich picture that could have affected the reliability of sugarcane supply. These are rainfall, harvesting schedules and haulage schedules. The extension services officer noted that some of the harvesting contractors in the area did not meet their schedules for reasons outside that of wet weather. This sort of behaviour could have had knock-on effects on hauler schedules as indicated in Figure 2. Moreover, the same officer noted that this inability to deliver on schedule was common with haulers that were contracted to "too many" growers. Delayed harvesting and haulage decrease the quality of sugarcane and consequently, affect economic returns. According to Reddy and Madhuri, harvest-to-crush delays can cause considerable moisture loss, sucrose inversion and a decline in recoverable sugar [120].

The issue of unfulfilled schedules may have been indicative of political issues. According to Ozkan-Tektas, the existence of calculative commitment other than affective commitment in a buyer-supplier relationship increases opportunism [121]. The "unfavourable" structure of harvesting contracts as indicated in Figure 2 may therefore, only had been a symptom of strained relationships between the contractors and the growers. Similarly, unfulfilled haulage schedules may have been indicative of poor relations. In their study, Gerwel-Proches and Bodhanya reported on conflicts between sugarcane haulers with both the millers and growers over unfulfilled schedules [122]. The extension services manager noted that some of harvesting contractors experienced frequent strike actions and that more often these labour issues affected harvesting schedules. Strikes have a tremendous cost to the employees as well as the entire supply chain.

The sugarcane laboratory manager indicated that some growers did not declare their consignment and field numbers on time (Figure 2). This caused problems for the supply office as in some instances the consignment and field numbers did not match. The issue of vehicle labelling, consignment and field numbers could be attributed to the practice of combining individual rateable deliveries for haulage purposes. This may also have been due to nonquota holders that use documents for quota growers. According to Simelane, there were cases within the milling area where growers planted in excess of their allocated quota (in nonquota land) [123]. Sugarcane grown on nonquota land is not monitored by the relevant authorities and as such, is more susceptible to pest and diseases. Collaboration provides leverage on culture and political forces (Figure 1), hence, enhanced collaboration between growers and both miller and harvesting contractors could possible provide leverage on vehicle labelling, consignment numbers, field numbers and the issue of unfulfilled harvesting schedules. Effective collaboration is based on shared values. Accordingly, higher levels of trust neutralises the negative effect of relative power and maintains shared values.

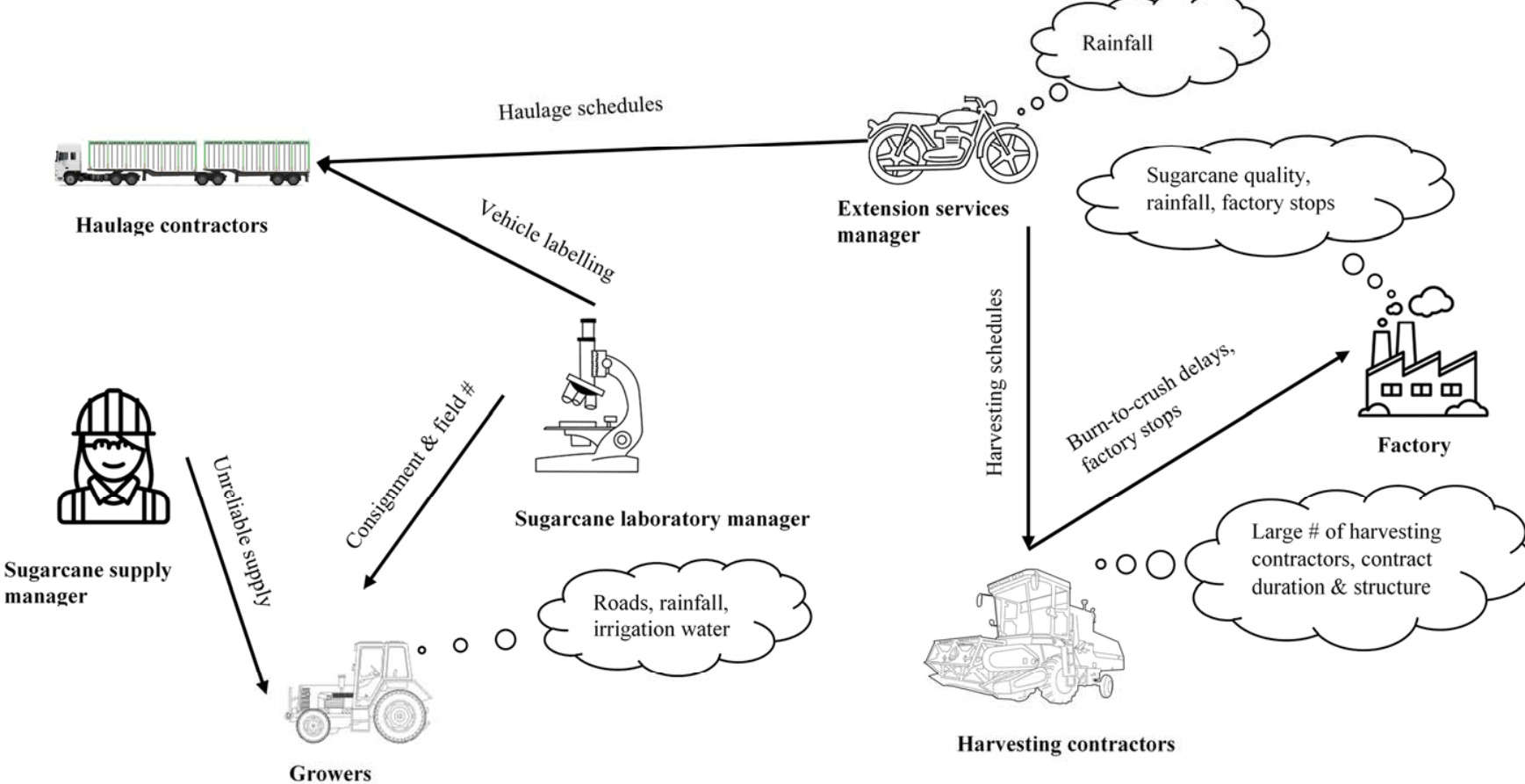

**Figure 2.** Rich picture diagram from stakeholder interviews.

The allocation of irrigation water to growers was mentioned as a constraint in the area. According to the grower (from one of the private farms), water allocation in the area was too bureaucratic. The grower stated that they (growers) used to obtain their water allocation through a now-defunct entity that acted between the growers and the main company that controls and distributes water. With the entity defunct, the grower noted that access to water had become too cumbersome. According to Howard, higher levels of bureaucracy often lead to water scarcity. Speelman is of the view that bureaucracy should be replaced by decentralised water allocation procedures that prioritise user participation. Access to water is one of the main criteria used to allocate sugarcane quota to growers. As such, all sugarcane grown in Swaziland is irrigated. Lack of irrigation as such could lead to the cancellation or nonrenewal of contracts and, consequently, economic hardships.

Unscheduled factory stops were also identified as a problem within the area (harvesting contractors and the factory manager). These stops include no-cane stops and shutdowns due to machine breakdown. According to the harvesting contractors, machine breakdowns were a source of most burn-to-crush delays in the area. This, in turn, caused the deterioration of sugarcane upstream. It was therefore not surprising that the factory manager also identified sugarcane quality as one of the main issues that drove inefficiencies in the area. Low recoverable sugar and poorly burnt sugarcane were some of the quality issues mentioned. Low recoverable sugar and consignments high in ash and fibre content were also identified by Hildbrand [124] in some milling areas in the South African ISSPS. An increase in green sugarcane correlates to an increase in machine breakdown as knives and shredders choke more frequently. Gomez et al. further reported that there is an increase in transport costs associated with green sugarcane as a result of lower trash densities [125].

The issue of sugarcane quality in the area could be attributed to burn-to-crush delays caused by machine breakdown, political issues and/or culture. Machine breakdown especially, early on in the season shifts most crushing towards the wet season (summer months). A combination of rainfall and high temperatures during this period, however, increases the rate of sugarcane deterioration. Most of the sugarcane harvested during the wet season is high in fibre [126]. Proper preburning of sugarcane at this time of the year is also compromised. Prolonged labour unrest on the other hand, directly impact scheduling (harvest and haulage) and, consequently, leads to the delivery of inferior quality sugarcane.

According to the diagnostic model (Figure 1), biophysical issues are correlated to both information sharing and economic issues. Hence, in the case of unscheduled factory stops (biophysical), information sharing and the economic domain are essential. Accurate information on factory stops is important in order to prevent the bullwhip effect. The sharing of accurate information on unscheduled stops is particularly important for the coordination of harvesting and haulage. A centralised information sharing policy with regard to unscheduled stops was important for the milling area, especially because the factory did not operate a sugarcane yard (stockpile). The cane payment system currently used in the area could also have indirectly contributed to these unscheduled stops (no cane). Cane payment in Swaziland is based on the polarisation percentage or pol % [127] and the weakness of this system is that growers do not fully bear the costs of extreme burn-to-crush delays. Sibomana et al. reported that some growers in the South African ISSPS deliberately delayed sugarcane delivery with the perception that this increases sucrose levels and consequently, returns [128].

The rich picture was well received at the meeting as participants approved its contents and further advised on other issues that were not captured by the earlier interviews. Besides understanding what was being portrayed by the rich picture, some of the stakeholders made fun of how they were represented on the picture. The sugarcane laboratory manager jokingly stated that "I can see myself under the microscope" whilst the sugarcane supply manager wondered "why are we represented by helmets". It was suggested after much deliberation that machine breakdown should be considered for further investigation. In a sugarcane mill, machine breakdowns are often associated with excessive soil and foreign matter. There is often an increase in foreign matter with rainfall. Accordingly, Moor posited that high mud levels often choke the shredder machine [129]. Machine breakdown in the factory could, therefore, be analysed along the different modes of failure, rainfall and maintenance. It is

also important to determine how and at what speed the breakdown information is communicated between stakeholders. This is critical because machine breakdowns have a ripple effect both up and downstream. The biophysical domain is interlinked to economic issues (Figure 1); hence, consideration of the economic impact of breakdowns on the system could provide some direction.

## 4. Conclusions

The adoption of technologies in IASPS is comparatively slow given the investment and potential benefits. The slow adoption is largely attributed to the complex nature of IASPS. In this research a systematic diagnostic model was developed and demonstrated at a sugarcane milling area. The model determines and evaluates interlinkages between the many domains, viz., biophysical, collaboration, culture, economics, environment, strategy, information sharing, structure and political forces. The model acts as a decision support mechanism to locate high leverage intervention points within and will also be used to make predictions the systems' behaviour. The model indicated that the collaboration domain, information sharing and supply chain structure had a higher direct leverage over the other domains as these were directly associated with a larger number of linkages. Collaboration and structure further provided dynamic leverage as these were part of feedback loops. The cultural domain was more predictive of collaboration compared to the other domains that were correlated to collaboration (information sharing, political forces and economics) as it had a relatively higher mean effect size. Similarly, structure was more predictive of information sharing compared to collaboration and the biophysical domain.

The model was demonstrated at a sugarcane milling area where issues that constrained productivity were identified. Results from the case study indicated that the issues that constrained productivity in the area were environmental (rainfall), biophysical (farm roads, factory stops, sugarcane quality and sugarcane delivery schedule), structural (irrigation water and harvesting contracts), political (grower infighting, harvesting schedules and haulage schedules), and cultural (labour unrest, vehicle labelling, consignment and field numbers). Using the IASPS diagnostic model, it was noted that most of the identified issues were linked to information sharing (factory stops, sugarcane supply) and collaboration (unfulfilled harvesting and haulage schedules, consignment and field numbers and sugarcane quality). Hence, collaboration and information sharing should be considered for any future interventions into the area as these could provide leverage to most of the constraints. Out of all the issues identified in the area, stakeholders suggested that machine breakdowns were the most serious constraint and that these should be investigated further. It is as such recommended that machine breakdown be analysed further especially along rainfall, preventative maintenance and information sharing.

Owing to the broad nature of some of the IASPS domains, the mean effect sizes (direction and magnitude) should be treated with caution as various constructs within each domain can have different effects within the same relationship. For example transaction costs and return on sale (economic domain) are negatively and positively correlated to collaboration, respectively. The limitation of the model is that articles for the meta-analysis were sourced from different industries, national conditions and economic environments. All these factors could have caused bias on the magnitude of the mean effect size. It is recommended that for future studies the model be continuously updated with linkages from other domain dimensions as these could provide a more holistic diagnosis.

**Author Contributions:** Conceptualization, M.I.S. and C.N.B.; Methodology, M.I.S.; Software, M.I.S.; Validation, M.S.S., T.S.W. and S.B.; Formal Analysis, M.I.S.; Investigation, M.I.S. and V.V.D.; Resources, M.I.S.; Data Curation, V.V.D.; Writing-Original Draft Preparation, M.I.S.; Writing-Review & Editing, V.V.D.; Visualization, M.I.S.; Supervision, C.N.B.; Project Administration, M.I.S.; Funding Acquisition, M.I.S.

**Funding:** This research received no external funding.

**Acknowledgments:** Our sincere gratitude is extended to Wadzanai Mafungu and Phyllis Kwenda for their invaluable assistance during data collection.

**Conflicts of Interest:** The authors declare no conflict of interest.

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
