# Peer review of "Developing a Systematic Diagnostic Model for Integrated Agricultural Supply and Processing Systems"

_systems, doi:10.3390/systems7010015_

Round 1

Reviewer 1 Report

This manuscript is up to a good shape, and it is not too far from publishable quality judging according to the scope and guidelines of the “Systems” journal. The following advice is given to improve the quality of this manuscript.

(1)    There are 10 hypotheses developed in the manuscript. However, they were not well defined considering all the factors are all well identified key factors to adoption while most of the null hypotheses were defined either correlated or related.  These hypotheses should be improved before considering being published, after all, it is such an obvious fact that these factors sit in a dynamic system and they are all related.

(2)    Many adoption factors summarized in Table 1 which require further defining, namely add notes to each of the factors to make sure all the factors hold consistent meaning.

(3)    The definitions of some of the adoption domains may be problematic, for example, “biophysical domain”, do the authors really mean “biophysical domain”?. Please double check it.

(4)     Multiple missing word/words, i.e. line 100 “A study by [36]…..”, was anything missing before the reference? Many typos like this across the whole manuscript. Please do serious proofreading.

(5)    The structure of the manuscript is not well balanced, in terms of the lengths of each section. It would be nice to improve it. 

Author Response

Reviewer comments are on the attached PDF file

Reviewer 2 Report

This was a well performed study, especially regarding the statistical components of the meta-analysis. There are some concerns regarding the soft systems methodology/qualitative component of the study. While the results of that portion were interesting, improvement to the description of both the process of soft systems methodology and how the data analysis was performed is warranted. While Checkland and Poulter's work is cited, the authors should include a description of the steps they offer, as well as a direct linkage to what the authors actually did in applying SSM with their study. 

Further, with the SSM data analysis, which commonly employs data analysis methods from other qualitative approaches, it is important to report a.) exactly how the analysis was performed (e.g. constant-comparative analysis, hermeneutics, critical ethnographic, etc.), b.)  who performed the analysis and their roles, and c.) how credibility and  trustworthiness of the findings was maintained (e.g. data triangulation, thick record development, multiple coders, reliability checks with developed codes, theme development, etc.) There should be a clear relationship among the data collected, analytic method, and outcomes. 

With the outcomes, it would be helpful to include quotes from participants that illustrate the findings, which also serves to support the claims made by the authors about the value of the rich pictures held by the stakeholders. The rich pictures themselves were a bit confusing and it was sometimes difficult to see a direct relationship between the visualization and the associated narrative, so that should be more clearly stated, possibly with numeric labeling in the figure that corresponds directly to portions of the narrative. 

It would also help to make clearer connections between the quantitative outcomes and the qualitative description, by explaining how you see the relation between the concepts (e.g. political, biophysical, etc.in all the qualitative sections which seems missing in places like lines 504-514. This will show a strong connection between the quantitative and qualitative outcomes you identify in the study and strengthen the value of the outcomes overall. 

Author Response

Reviewer comments are on attached PDF
